# Deaf intermarriage has limited effect on the prevalence of recessive deafness and no effect on underlying allelic frequency

Derek C. Braun[1]*, Samir Jain[1], Eric Epstein[1], Brian H. Greenwald[2‡], Brienna Herold[1‡], Margaret Gray[1‡]

1 Department of Science, Technology, and Mathematics, Gallaudet University, Washington, D.C., United States of America, 2 Department of History, Philosophy, Religion, and Sociology, Gallaudet University, Washington, D.C., United States of America

☯ These authors contributed equally to this work.
‡ These authors also contributed equally to this work.
* derek.braun@gallaudet.edu

## Abstract

The idea that deaf intermarriage increases the prevalence of deafness was forcefully pushed in the late 19th century by Alexander Graham Bell, in proceedings published by the National Academy of Science. Bell's hypothesis was not supported by a 19th century study by Edward Allen Fay, which was funded by Bell's own organization, the Volta Bureau. The Fay study showed through an analysis of 4,471 deaf marriages that the chances of having deaf children did not increase significantly when both parents were deaf. In light of an apparent increase in non-complementary pairings when a modern dataset of Gallaudet alumni was compared with the 19th century Fay dataset, Bell's argument has been resurrected. This hypothesis is that residential schools for the deaf, which concentrate signing deaf individuals together, have promoted assortative mating, which in turn has increased the prevalence of recessive deafness and also the commonest underlying deafness allele. Because this hypothesis persists, even though it contradicts classical models of assortative mating, it is critically important that it be thoroughly investigated. In this study, we used an established forward-time genetics simulator with parameters and measurements collected from the published literature. Compared to mathematical equations, simulations allowed for more complex modeling, operated without assumptions of parametricity, and captured ending distributions and variances. Our simulation results affirm predictions from classical equations and show that intense assortative mating only modestly increases the prevalence of deafness, with this effect mostly completed by the third generation. More importantly, our data show that even intense assortative mating does not affect the frequency of the underlying alleles under reported conditions. These results are not locus-specific and are generalizable to other forms of recessive deafness. We explain the higher rate of non-complementary pairings measured in the contemporary Gallaudet alumni sample as compared to the Fay dataset.

**Data Availability Statement:** The source code and dataset created for this study are publicly available from https://github.com/derekbraun/homogamy.git

so that anyone can replicate our experiments and build upon our work.

**Funding:** The author(s) received no specific funding for this work.

**Competing interests:** The authors have declared that no competing interests exist.

## Introduction

In an 1883 presentation to the National Academy of Sciences, Alexander Graham Bell delivered an ominous warning about the intermarriage of deaf individuals [1]. If intermarriage was left unchecked, Bell argued, this would lead to a "deaf variety of the human race." Bell delineated the costs of educating deaf individuals and argued that the residential schools were an economic burden to state governments funding the schools [1]. Following this address, Bell conducted research on hereditary deafness on Martha's Vineyard in the late 1880s. Bell persisted in efforts to better understand the transmission of genetic deafness, although he ultimately never understood it [2, 3]. To this end, he hired Edward Allen Fay, who was the vice-president at Gallaudet College and editor of the American Annals of the Deaf. Bell's Volta Bureau funded Fay's landmark study [4] of 4,471 deaf marriage pedigrees, collected from alumni of Gallaudet, whose students were deaf, and alumni from residential schools for the deaf throughout the United States. Although the Fay study eventually concluded that deaf intermarriage did not much increase the chances of having deaf children [4], Bell remained vocal in his beliefs that the ideal marriage was a marriage between a deaf and hearing person. As the wealthy inventor of the telephone, he remained highly influential in the scientific community and in the nascent eugenics movement.

The results of Fay's study were not well understood at the time because they were published in 1898, before the rediscovery of Mendel's work [5] and the many discoveries in experimental and theoretical genetics that followed. In those days, the understanding of the heredity of deafness and heredity in general was based on observation, such as those recorded by the otologist William Wilde in 1857, or by tallying summary statistics, like Fay's work [6].

We now know that genetic deafness [OMIM 220290] accounts for the majority of deafness in children and is caused by mutations in >140 already mapped genes [7]. Of these, genetic deafness due to recessive connexin 26 (*GJB2*) variant alleles [OMIM 121011] is by far the most common, accounting for more than a quarter of congenital deafness [8]. Three *GJB2* frameshift variants account for most severe to profound congenital deafness, and are associated with specific ethnic groups: c.35delG in European ancestry, c.167delT in Ashkenazi Jewish ancestry, and c.235delC in Asian ancestry [9–11]. These three variants are inherited in an autosomal recessive fashion, and cause nonsyndromic deafness, meaning that there are no other discernible physical characteristics.

In fact, most deaf intermarriage produces hearing children mainly because of the complementation between the many recessive genes causing deafness, and also because deafness often occurred or was observed after birth and was usually attributed to injury or childhood morbidity. Fay [4] termed these cases as "adventitious."

In the early 20th century, several authors, but most notably R.A. Fisher [12] and Sewell Wright [13] wrote classical theoretical essays on assortative mating. Their theory and its mathematics were later reviewed and reworked by Crow and Felsenstein [14]. As a simple theoretical example, if deaf people sought out one another and only mated with one another, and if all deaf people also all carried the same recessive allele, then every mating of a deaf pair would result in only deaf children. The mating of hearing individuals, of whom a small proportion are heterozygote carriers, would result in the occasional deaf child, who would then grow up to seek a deaf mate, joining the deaf subpopulation. Over several generations, this would increase homozygosity in the population, and consequently, also increase the prevalence of deafness. However, even though phenotypic expression of deafness would have increased, the frequency of that underlying recessive deafness allele would remain the same in the population: underlying recessive alleles would have simply moved from the hearing subpopulation into the deaf subpopulation. The hearing subpopulation would be made up of fewer

heterozygote carriers. In summary, the classical model asserts that assortative mating increases homozygosity and the phenotypic expression of recessive alleles, but the underlying allelic frequencies in the population do not change. For further reading on the theoretical effects of assortative mating, with an example of the deaf community, we refer readers to Crow and Felsenstein [14].

In contradiction to the classical model, recent authors have posited that assortative mating between signing deaf individuals who socialized together in residential schools over slightly more than 200 years in the United States, has increased the prevalence of recessive deafness, and have claimed that it has also increased the frequency of the underlying deafness allele(s) [15, 16]. Termed "linguistic homogamy," this is reasoned to be motivated by an innate human need for easy and effective communication. Signing deaf individuals would find easy communication with one another and be motivated to intermarry. This hypothesis was used to explain results from a pedigree study by Arnos *et al*. [17] which had shown that non-complementary pairings in a contemporary Gallaudet alumni dataset were measured at a higher rate than non-complementary pairings in the original Fay dataset, which had been collected more than 100 years earlier.

The degree of assortative mating among deaf Americans has been measured several times, and has remained relatively stable over the past 200 years. Fay [4], in 1898, initially reported an uncorrected measure of 72.5%. From the 1970 National Census of the Deaf Population (NCDP) in the United States, Schein and Delk [18] calculated a figure of 80–90%. Most recently. Blanton *et al*. [19] calculated a figure of 79% from a sample of Gallaudet alumni, who are predominantly white Americans.

The reproductive fitness of deaf individuals, as measured by fertility, has also been investigated. The literature reports markedly depressed relative fitness. Values normalized against the general population range from 0.31 to 0.91. The highest measured fitness of 0.91, which is still depressed, was reported from an educated American deaf sample from Gallaudet [18–22].

This continuing question of whether assortative mating between deaf individuals affects the prevalence of recessive deafness and the frequency of the underlying alleles is important because it has potential implications for policy and funding decisions. This includes popular support for funding of residential deaf education programs, which bring deaf people together into social groups. It may also alter opinions about eugenics, which was popular in recent history, particularly in Germany and in Scandinavian countries. Discussion about the ethics of eugenics is moving to the forefront again, given the recent use of gene-editing technologies (CRISPR) in humans [23, 24]. It is therefore critically important that the question "does deaf intermarriage (assortative mating) increase the prevalence of deafness and the underlying alleles" is carefully examined using a variety of approaches.

In this study, we performed thousands of forward-time evolutionary simulations using an established package [25, 26]. We simulated assortative mating and measured the changes in deafness and the underlying allelic frequencies, using parameters scoured from the published literature. We used statistical analyses to compare ending distributions of phenotypic deafness and allele frequencies. This approach allowed us to capture the variance in end results without any underlying assumptions of parametricity. We further compared these results with mathematical modeling.

## Results

We initially ran 5,000 simulations following parameters established by Nance and Kearsey [15] so that we could directly compare our simulation results with theirs. First, the initial frequency of the simulated recessive deafness allele was set to 1.304%. This frequency is approximately at

the midpoint of the 0.6% to 3.5% reported range of carrier frequencies for the c.35delG variant of *GJB2* in white Americans and Europeans [9, 27, 28]. Simulations were run over 20 generations (400 years), which reflects the approximate time frame among white Americans and Europeans that signed languages are believed to have existed, deaf individuals have formed close social ties, and assortative mating among deaf individuals has been occurring [15]. A constant population size of 200,000 simulated individuals was set [15]. At each generation, a proportion of simulated hearing individuals were randomly selected (of which a small proportion, by random choice, carried a single recessive deafness allele), and assigned the deafness phenotype at a conservative rate of 0.8 per 1,000 simulated individuals, which is a measured frequency of profound deafness at birth [29, 30]. This assigned deafness, biologically, reflects genetic deafness due to the many other complementary genes which cause deafness, deafness from epigenetic causes, and/or perinatal morbidity. Although the prevalence of identified deafness continues to increase throughout childhood to approximately 3.5 per 1,000 [30], this higher figure was not used in our simulations. Genetically deaf simulated individuals and simulated individuals with assigned deafness (from other causes) were mated together in the same pool. For our initial analysis, assortative mating was set to 0% and 90% to create two datasets for endpoint comparison; in subsequent analyses, simulations were run over a range of degrees of assortative mating. The 90% assortative mating figure is per Nance and Kearsey [15]; see Background for further detail.

Under these parameters, after 20 generations (400 years), the median frequency of deaf individuals with our recessive allele increased by 23% relative to the simulations with no assortative mating, that is, 0.017% as compared to 0.0220%, which is statistically significant (Fig 1 and Table 1; *n* = 5,000 simulations, Mann-Whitney *U* = 5.83 x 10^6, $p < 10^{-308}$, common language effect size *f* = 76.68%). This statistic was identical to the calculation of 0.0220% using equation (3) from Crow and Felsenstein [14] (Table 1) and described in Materials and Methods. Most of the change occurred within the first three generations. Notably, this figure is 7-fold less than the ~0.16% frequency reported elsewhere in a comparable simulation of deaf-deaf assortative mating with essentially the same parameters [15].

The frequency of the recessive deafness allele did not increase significantly; it was 1.304% versus 1.306% after 20 generations (Fig 1 and Table 1; *n* = 5,000 simulations, Mann-Whitney *U* = 1.25 x 10^7, *p* = 0.94, common language effect size *f* = 49.96%). Likewise, this figure is also much less than the ~1.7% frequency reported elsewhere in a simulation with essentially the same parameters [15].

The inbreeding coefficient, *F*, was different: 0 versus 0.00376, which was statistically significant (*n* = 5,000 simulations, Mann-Whitney *U* = 4.98 x 10^6, $p < 10^{-308}$, common language effect size *f* = 80.09%. This increase was small since *F* was being attenuated by competition for mates between the small number of simulated genetically deaf individuals and the larger pool of simulated individuals with assigned deafness due to other causes (*e.g.* complementary genes).

We next ran simulations over a range of degrees of assortative mating (Fig 2 and Table 1). The prevalence of deafness increased proportionately to assortative mating. Most of the change occurred in the first three generations. All differences in the prevalence of deafness between each endpoint were highly significant with all $p < 10^{-40}$. The results were also in close agreement with calculations using equation (3) from Crow and Felsenstein [14] (Table 1) and described in Materials and Methods. The frequency of the underlying recessive allele, however, remained invariable regardless of the extent of assortative mating, and no statistically significant difference was found between any of the endpoints (*n* = 5,000 simulations, Kruskal-Wallis *H* = 1.5, *p* = 0.69, effect size $\eta^2 \approx 0.0\%$).

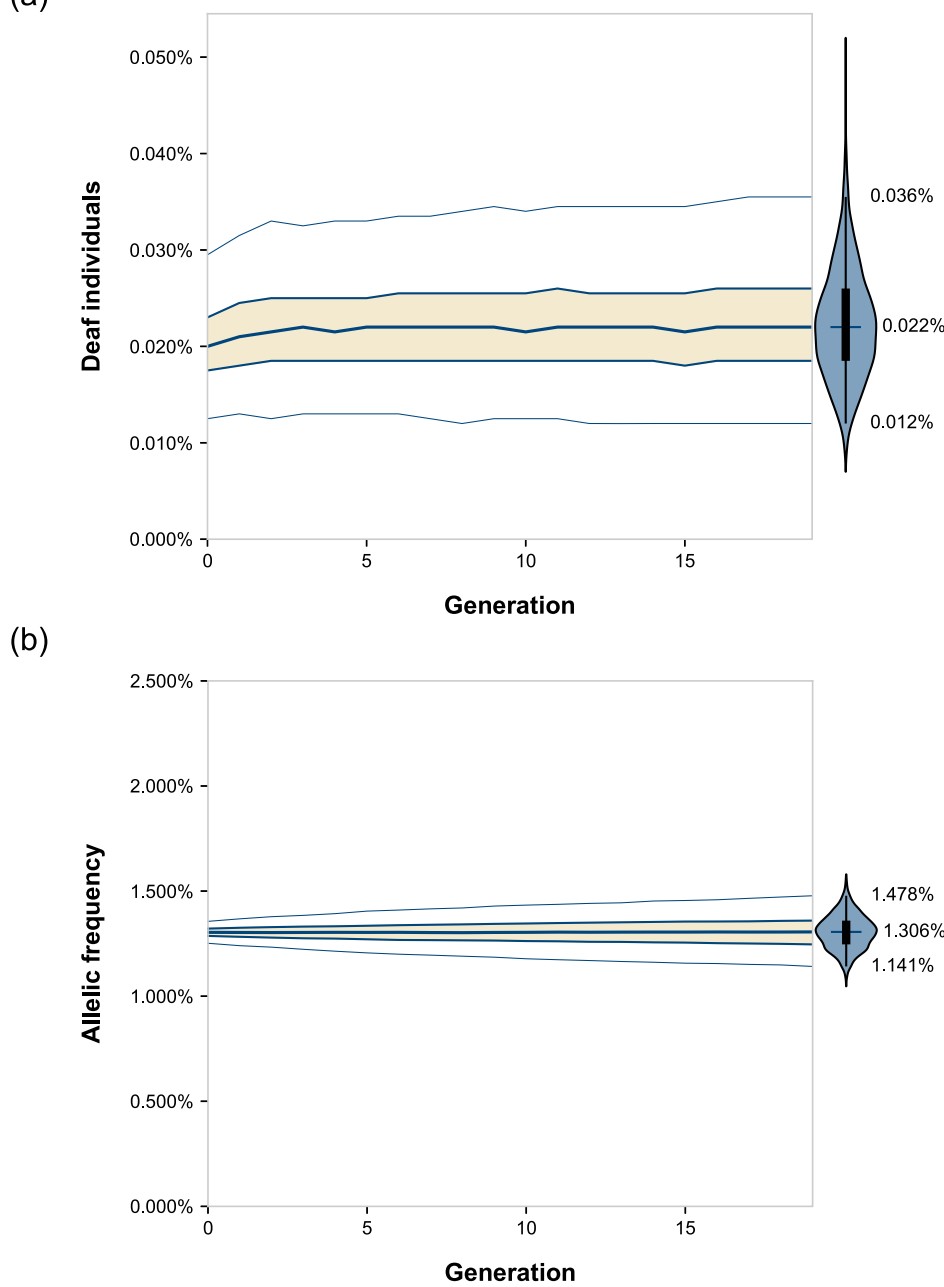

**Fig 1. Effect over time of assortative mating on the prevalence of deafness and the frequency of the underlying recessive deafness allele.** Lines, from top to bottom, represent a five-number summary: 98% percentile, 75% quartile, median, 25% quartile, and 2% percentile. To the right of each subplot is a violin plot showing the distribution of the endpoint data. The tips of the violins represent the extrema. The vertical lines within the violins show the 2% through 98% percentile. The boxes within the violins show the first through third quartile. The cross-hatches show the medians. Simulations were run with relative fitness = 1.0 and other parameters as described in Materials and Methods.

## Synergistic effects of fitness and assortative mating on allelic frequency

Because assortative mating increases the phenotypic expression of alleles, it would therefore modulate the effects of selective pressure upon those alleles. Nance and Kearsey [15] have argued that "relaxed" fitness would facilitate increasing the numbers of deaf individuals. We

**Table 1. Simulation results showing effects of assortative mating and fitness on the frequencies of deafness and the underlying recessive allele after 200 years.**

Deaf Individuals

| | | Degree of Assortative Mating | | | |
| --- | --- | --- | --- | --- | --- |
| | | 0% | 30% | 60% | 90% |
| **Genetic Fitness** | 0.0 | 0.0105% (0.0050–0.0175%) | 0.0105% (0.0055–0.0175%) | 0.0105% (0.0055–0.0170%) | 0.0105% (0.0050–0.0175%) |
| | 0.5 | 0.0130% (0.0070–0.0210%) | 0.0135% (0.0070–0.0215%) | 0.0135% (0.0075–0.0220%) | 0.0140% (0.0075–0.0225%) |
| | 1.0 | 0.0170% (0.0100–0.0260%) | 0.0180% (0.0105–0.0280%) | 0.0195% (0.0110–0.0310%) | 0.0220% (0.0125–0.0350%) |
| | | **0.0170%**[*] | **0.0182%**[*] | **0.0198%**[*] | **0.0220%**[*] |
| | 1.5 | 0.0220% (0.0135–0.0330%) | 0.0260% (0.0155–0.0400%) | 0.0345% (0.0190–0.0575%) | 0.0770% (0.0310–0.284%) |
| | 2.0 | 0.0305% (0.0195–0.0440%) | 0.0445% (0.0265–0.0690%) | 0.1485% (0.0565–0.4425%) | 29.20% (3.43–64.1%) |

Allelic Frequency

| | | Degree of Assortative Mating | | | |
| --- | --- | --- | --- | --- | --- |
| | | 0% | 30% | 60% | 90% |
| **Genetic Fitness** | 0.0 | 1.03% (0.908–1.16%) | 1.03% (0.912–1.16%) | 1.03% (0.910–1.16%) | 1.03% (0.909–1.16%) |
| | 0.5 | 1.15% (1.02–1.30%) | 1.15% (1.01–1.30%) | 1.15% (1.01–1.29%) | 1.14% (1.00–1.28%) |
| | 1.0 | 1.30% (1.15–1.47%) | 1.30% (1.15–1.47%) | 1.30% (1.15–1.47%) | 1.31% (1.15–1.47%) |
| | 1.5 | 1.49% (1.31–1.70%) | 1.52% (1.33–1.74%) | 1.56% (1.35–1.81%) | 1.70% (1.41–2.22%) |
| | 2.0 | 1.74% (1.52–2.00%) | 1.86% (1.60–2.17%) | 2.32% (1.78–3.41%) | 36.4% (6.17–73.3%) |

Values given are medians, with 2% through 98% percentiles in parentheses. Values in bold and followed by an asterisk were calculated from equation (3) from Crow and Felsenstein [14] as described in Materials and Methods. Simulations were run as described in Materials and Methods.

therefore simulated assortative mating across a range of relative fitnesses. We show that the frequency of the underlying recessive deafness allele was sensitive to relative fitness, which became noticeable at or above fitnesses of 1.5 when combined with assortative mating. (Fig 3 and Table 1).

## Discussion

In this study, we addressed the century-old debate about whether deaf intermarriage increases the prevalence of deafness. We also investigated a recent claim, based on simulations, that assortative mating could increase the underlying recessive allelic frequency [15, 16]. We ran forward-time computer simulations using the `simuPOP` package, which has been used by others for assortative mating simulations, to test the hypothesis that assortative mating among deaf Americans (deaf intermarriage; linguistic homogamy) would affect the prevalence of deafness as well as the underlying recessive allelic frequencies [25, 26]. For each scenario, we analyzed the results of 5,000 simulations. We used statistical analyses to compare ending distributions. We used this approach in addition to mathematical modeling because it allowed us to capture the variance in results without underlying assumptions of parametricity, and allowed us to easily set up the more complex scenario of introducing individuals with acquired or complementary deafness to the mating pool.

Our simulations confirm that intense (90%) assortative mating would increase the prevalence of deafness by 23% over 20 generations. Most of this increase would occur within the first three generations of assortative mating. These results were in nearly identical agreement with mathematical predictions using Crow and Felsenstein's [14] equation (3) (Table 1). The theoretical maximum amount that the prevalence of phenotypic deafness could increase by is significant, but when we used real-world data to run our simulations, we showed that this increase is quite limited. It's limited because of the many different, complementary genes that can cause deafness, as well as non-genetic causes for deafness.

(a)

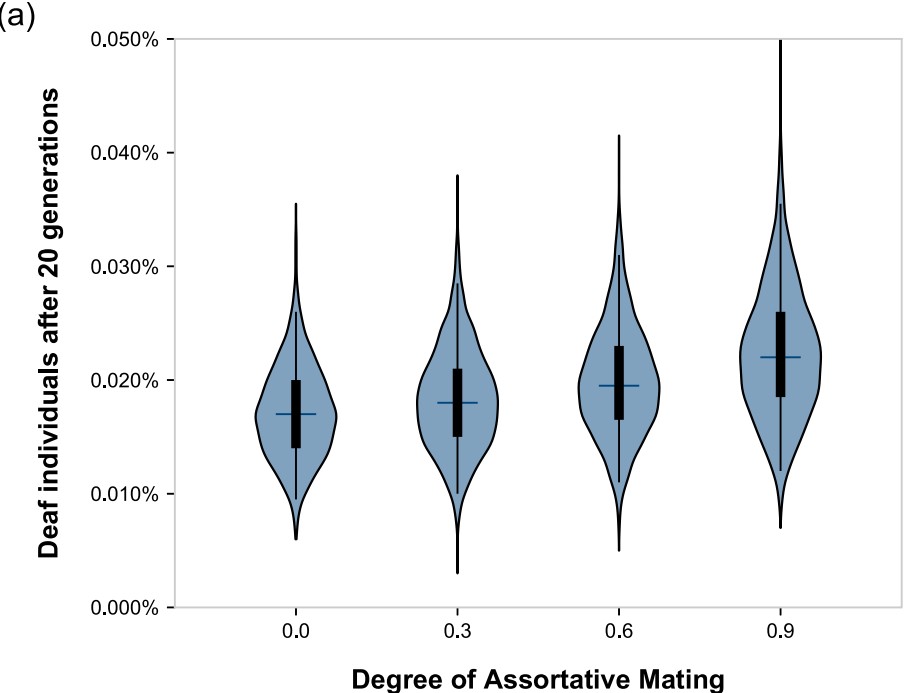

(b)

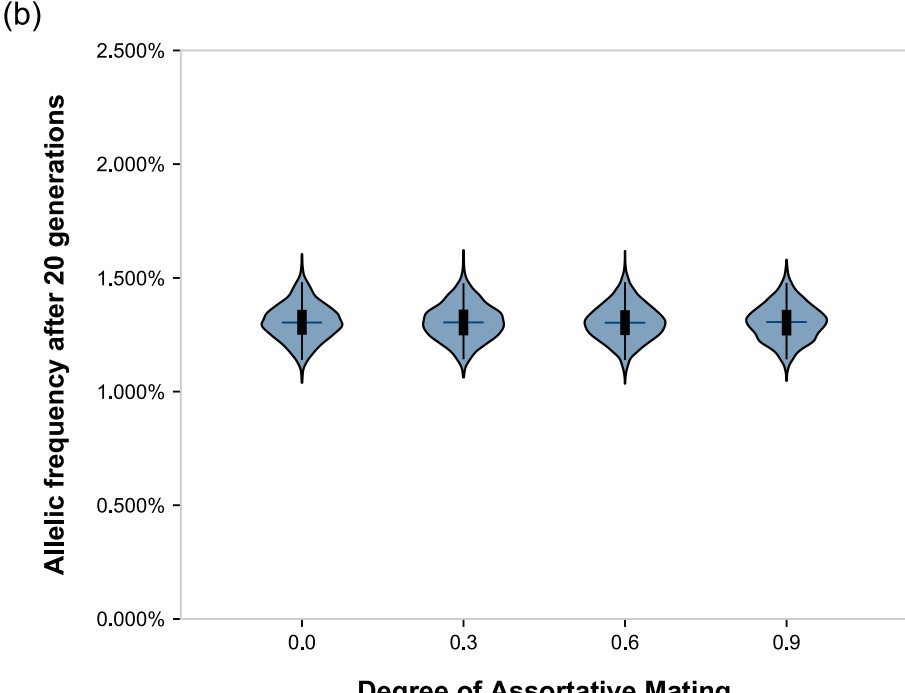

**Fig 2. Effect of different degrees of assortative mating on the prevalence of deafness and the frequency of the underlying recessive deafness allele.** Violin plots show the distributions of the endpoint data after 20 generations. The tips of the violins represent the extrema. The vertical lines within the violins show the 2% through 98% percentile. The boxes within the violins show the first through third quartile. The cross-hatches show the medians. Simulations were run with relative fitness = 1.0 and other parameters as described in Materials and Methods.

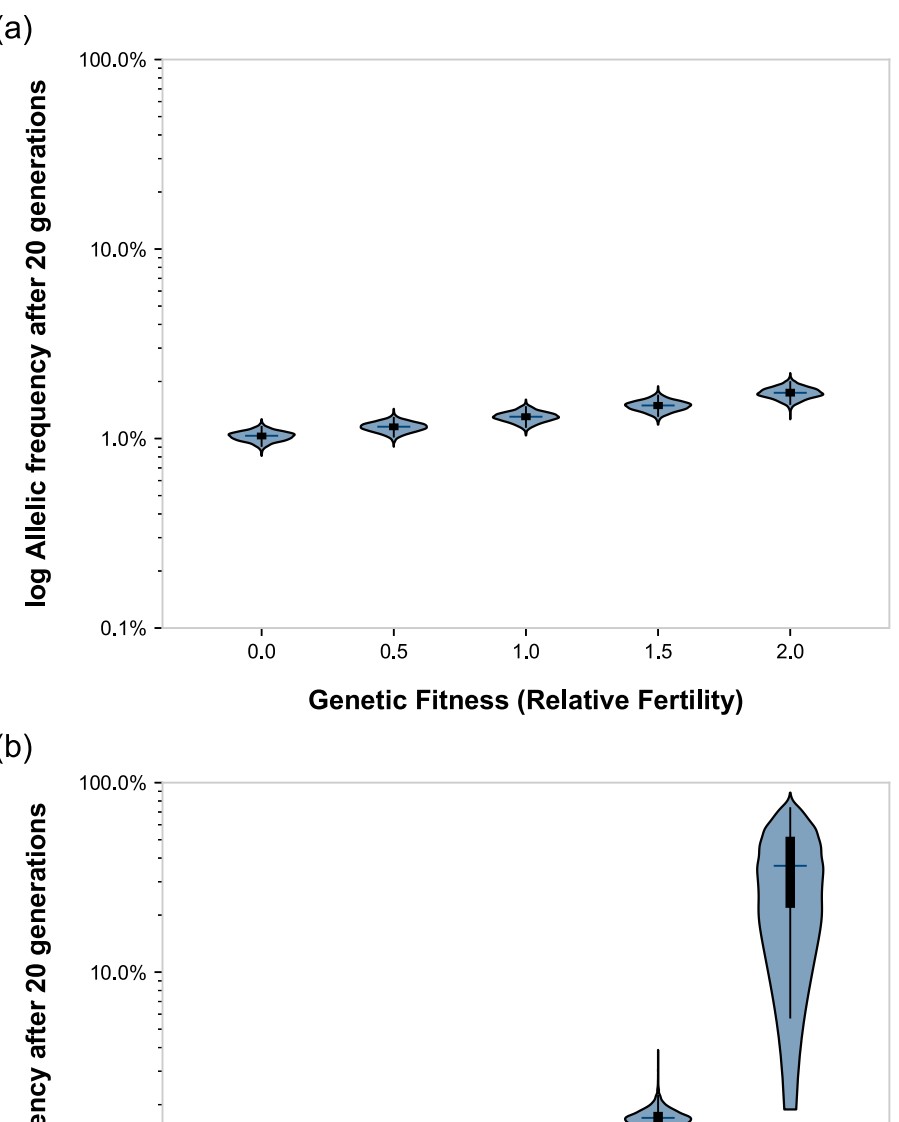

**Fig 3. Synergy of assortative mating and fitness on the prevalence of deafness and the frequency of the underlying recessive deafness allele.** Right: no assortative mating; left: 90% assortative mating. Violin plots show the distributions of the endpoint data after 20 generations. The tips of the violins represent the extrema. The vertical lines within the violins show the 2% through 98% percentile. The boxes within the violins show the first through third quartile. The cross-hatches show the medians.

Importantly, the simulations also confirmed that assortative mating did not affect the underlying allelic frequencies at all (Figs 1 and 2 and Table 1). Our results did not match with the results of a computer simulation published elsewhere using essentially the same parameters, which predicted a ~700% increase in the prevalence of deaf individuals, as well as a ~30% increase in the frequency of the recessive deafness allele [15].

Our simulation results also confirm that relative reproductive fitness impacts allelic frequencies. Because assortative mating increases the phenotypic expression of rare alleles, relative fitness acted synergistically with assortative mating in our simulations to accelerate changes in underlying recessive allelic frequencies. In our simulations with greatly exaggerated relative fitness (1.5x and 2x), the frequency of the underlying recessive deafness allele increased. However, these results do not apply to the worldwide deaf community today because there are no reports in the literature of any group of deaf individuals with higher-than-normal relative fitness. Instead, the literature uniformly reports *depressed* relative fitness, ranging from 0.31 to 0.91; the highest measured fitness of 0.91, which is still depressed, was from an educated American deaf sample from Gallaudet [18–22]. Therefore, in real-world application, the synergy of assortative mating with depressed fitness should lightly *constrain* the frequency of recessive deafness alleles at low levels (Fig 3 and Table 1).

Nance [15] forwarded a hypothesis that assortative mating based on shared language compatibility, which he termed "linguistic homogamy," among early humans may have accelerated the evolution of human speech genes, particularly *FOXP2*, some 150,000 years ago. Nance argued that improved language skills in early humans likely correlated with better cooperation, better survival and higher fitness. Our simulation results support this intriguing hypothesis. Preferential, assortative mating between those with advantageous language capabilities, who therefore have higher fitness, could have synergistically accelerated the evolution of speech and language.

In 2008, Arnos and colleagues [17] studied contemporary pedigrees collected from Gallaudet alumni, and compared them with pedigrees from Fay's study, which were ascertained from Gallaudet alumni and deaf institutions across the country. Segregation analysis comparing these two datasets showed that the proportion of non-complementary pairings were 4.2% in the 1801–1899 Fay dataset and 23% in the 2007 Gallaudet alumni dataset. This difference in non-complementary pairing shows more homozygosity in the modern sample. These figures agree with our data, collected from the literature, from which we calculated that homozygosity for the 35delG variant of *GJB2* in the American deaf community must currently be around 21%—by dividing the two statistics of 0.017 per 1,000 who are homozygous for the 35delG *GJB2* frameshift by the 0.08 per 1,000 who are born deaf [15, 29]. Importantly, this measured increase in non-complementary pairings only reflects increased homozygosity and has no bearing on the frequency of the underlying alleles in the population. Based on our simulation results, we do not expect this homozygosity (and the prevalence of deafness) to continue to increase since we have surpassed the third generation of assortative mating.

We are left with the puzzling paradox of how the commonest *GJB2* variant alleles causing severe to profound deafness: c.35delG, c.167delT, and c.235delC, have been measured at frequencies of between 1% and 4.4%, while measurements of reproductive fitness in deaf communities have been uniformly depressed [9–11, 18–22]. These three frameshift alleles account for the majority of severe to profound nonsyndromic deafness in white Americans [9–11]. One possibility is mutation-selection equilibrium: novel *GJB2* mutations are perhaps being introduced at the same rate that mutations in the gene pool are being eliminated. Evidence showing a mutational hotspot at *GJB2*, particularly for deletion mutations, would provide support for this hypothesis. A second, and intriguing possibility is that of balancing selection. Unrelated to studying deafness, Tran van Nhieu, Clair *et al.* [31] have shown in tissue culture experiments

that *Shigella flexneri* requires *GJB2* for egression into the intestinal epithelia, raising the possibility that the three common *GJB2* deletions could confer resistance to dysentery.

Connexons are dimers of hexameric proteins made up of individual connexins; in individuals with *GJB2* deletions, *GJB2* is replaced by other connexins to form connexons which appear to retain normal function everywhere except for the cochlea. Dysentery has been endemic at least since the advent of urbanization, and resistance to this disease via altered connexons may have provided enough positive selection to bring the commonest *GJB2* mutations to their present frequencies. This hypothesis is intriguing and should be investigated. Further, it would be interesting to see if this advantage exists only for *GJB2* variant homozygotes, or if heterozygous carriers for recessive *GJB2* deafness would also be resistant to shigellosis.

We hope that this study can put to rest the century-long argument put forth by Alexander Graham Bell [1] that deaf intermarriage would lead to a "deaf variety of the human race." Using simulations and drawing upon mathematical modeling, with measurements and parameters collected from the published literature over more than a century of data, our results unequivocally affirm the models for assortative mating famously credited to R.A. Fisher [12] and Sewell Wright [13]. That is, our data show that while deaf intermarriage initially had some effect on the prevalence of deafness, this effect was limited and mostly completed by the third generation of assortative mating. However, deaf intermarriage and assortative mating did not, and should not, change the frequency of underlying recessive deafness alleles, unless and until there is strong positive selection present. But there isn't strong positive selection present. Therefore, Alexander Graham Bell's [1] concerns about a "deaf variety of the human race" will not happen even if deaf intermarriage and assortative mating continue its present course.

## Materials and methods

### Code and dataset

The source code and dataset created for this study are publicly available from https://github.com/derekbraun/homogamy.git so that anyone can replicate our experiments and build upon our work.

### Simulations

Simulations were performed using `simuPOP 1.1.10.8` which is a forward-time population genetics package, scriptable via Python, that has been used to simulate assortative mating [25, 26]. Simulations were scripted with `Python 3.7.4` on a computer running `macOS 10.14.6`. Simulations were parallelized on a 16-core Intel Xeon workstation. It required 80 hours of CPU time to complete the final simulations shown in this manuscript. We modeled both assortative mating (homogamy) and reproductive fitness using a non-monogamous mating scheme. Non-monogamous mating was chosen, after some experimentation with code, because this allowed for better stability in the final proportion of assortative mating per generation given the small number of deaf individuals in the simulated population. Sexes were not assigned to individuals; this was decided, after some experimentation with code, because it simplified coding and sped up execution time.

After each generation, the following was calculated: the frequencies of the dominant and recessive alleles A and a; the frequencies of the homozygous dominant, heterozygous, and homozygous recessive genotypes AA, Aa, and aa; the number of individuals with each genotype; the number and frequency of deaf individuals (including acquired deafness); and the

inbreeding coefficient ($F$) calculated as follows:

$$F = \begin{cases} 1 - \dfrac{P(Aa)}{2pq}, P(Aa) \leq 2pq \\ 0, otherwise \end{cases}$$

The frequency of deafness alleles from simulations were also compared to those calculated from equation (3) of Crow & Felsenstein [14]. The effective assortative mating fraction, $r$, was derived from the % assortative mating (homogamy) and re-estimated after each generation by adjusting it by the size of the mating pool. This calculation matches the logic used in the forward simulation script which is that the initial mating pool size included all forms of profound deafness at the rate of 0.8 per 1,000 individuals [29]. Therefore, initially and before assortative mating, $q^2$ individuals have genetic deafness due to connexin 26, and $0.008 - q^2$ individuals have acquired or complementary genetic deafness. At $t_0$, $R_t = q^2$, so at $t_0$, the expression for the mating pool size, $0.008 - q^2 + R_t$ simplifies to just 0.008. As assortative mating progresses in successive generations, $R_t$ increases, and the mating pool size becomes slightly larger, as follows:

$$P(aa) = R_{t+1} = (1-r)q^2 + r\left[\frac{q^2 + R_t(p-q)}{1-R_t}\right], r = homogamy\,\%\left[\frac{R_t}{0.0008 - q^2 + R_t}\right]$$

## Statistical testing and graphing

Statistical comparisons between datasets were performed using `SciPy 1.3.0`. We performed the Shapiro-Wilk test of normality on ending frequencies. Since these ending frequencies were often not normally distributed, and because we additionally wished to test for significant differences in both medians and variances, we used nonparametric tests: the Mann-Whitney $U$ test for two independent groups or the Kruskal-Wallis test for $k$ independent groups. Significant Kruskal-Wallis $p$-values were followed by *post hoc* pairwise Mann-Whitney tests without Bonferroni correction, which is more sensitive than Dunn's test [32]. Figures were generated in Python using `matplotlib 3.1.1`.

## Acknowledgments

We acknowledge Trevor Klemp and Ashley Bergeron for their help on this project during their summer internships. We thank Mohammad Obiedat and Gaurav Arora for critical comments on drafts.

## Author Contributions

**Conceptualization:** Derek C. Braun, Brienna Herold.

**Data curation:** Derek C. Braun, Samir Jain.

**Formal analysis:** Derek C. Braun, Samir Jain, Eric Epstein.

**Investigation:** Derek C. Braun.

**Methodology:** Derek C. Braun, Samir Jain, Eric Epstein, Margaret Gray.

**Project administration:** Derek C. Braun.

**Resources:** Derek C. Braun, Brian H. Greenwald.

**Software:** Derek C. Braun, Samir Jain, Eric Epstein, Margaret Gray.

**Supervision:** Derek C. Braun.

**Validation:** Derek C. Braun.

**Visualization:** Derek C. Braun.

**Writing – original draft:** Derek C. Braun, Eric Epstein, Brian H. Greenwald.

**Writing – review & editing:** Derek C. Braun, Samir Jain, Eric Epstein, Brian H. Greenwald, Brienna Herold.

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
