## [Decision Letter · Decision Letter 0]

24 Jun 2020

PONE-D-20-07542

Deaf intermarriage does not increase the prevalence of deafness alleles

PLOS ONE

Dear Dr. Braun,

Thank you for submitting your manuscript to PLOS ONE. We apologize for the lengthy review process that was due to these challenging times. After careful consideration, we feel that your manuscript has merit but does not fully meet PLOS ONE’s publication criteria as it currently stands.

The reviewers have raised serious concerns regarding the statistical analyses and the inclusion of only a limited subset of pathogenic variants in *GJB2* which could impact the accuracy of the data and consequently the conclusions. Therefore, we invite you to submit a revised version of the manuscript that addresses the points raised during the review process.

We look forward to receiving your revised manuscript.

Kind regards,

Hela Azaiez, Ph.D

Academic Editor

PLOS ONE

Journal Requirements:

"We thank the NASA/D.C. Space Grant Consortium and the Beverly Taylor Sorensen Student Fellowship for their generous support of the student interns who worked on this project."

Reviewers' comments:

Reviewer's Responses to Questions

**Comments to the Author**

1. Is the manuscript technically sound, and do the data support the conclusions?

Reviewer #1: Partly

Reviewer #2: Yes

2. Has the statistical analysis been performed appropriately and rigorously? 

Reviewer #1: I Don't Know

Reviewer #2: No

3. Have the authors made all data underlying the findings in their manuscript fully available?

Reviewer #1: No

Reviewer #2: Yes

4. Is the manuscript presented in an intelligible fashion and written in standard English?

Reviewer #1: Yes

Reviewer #2: Yes

5. Review Comments to the Author

Reviewer #1: Braun et al perform a simulation analysis to examine the effects of assertively matting of deaf individuals on overall allele prevalence and phenotypic prevalence. Prior to this study, data suggested that deaf-deaf mating would increase the rate of deafness. However, based on the authors simulations and applied assumptions, the data suggests this is not true.

Overall the manuscript is well written.

The manuscript could be improved by revisions.

The introduction should be trimmed. There is a lot of superfluous information that isn’t required.

Lines 91-93: “Three GJB2 frameshift variants account for nearly all GJB2 deafness, and are associated with specific ethnic groups: 35delG in European ancestry, 167delT in Ashkenazi Jewish ancestry, and 235delC in Asian ancestry [12-14]” – This is not true. The p.M34T and p.V37I are the most common contributors to GJB2-related hearing loss. The frameshifts the authors mention account for most of the severe to profound hearing loss cases associated with GJB2-related deafness. Please revise this.

Please refer to the variants using the proper HGVS cDNA notation (ie “35delG” to “c.35delG”).

Lines 97-98: It is unclear what the authors mean by “acquired” deafness. Just because the deafness is not congenital does not mean it does not have a genetic origin. Many forms of autosomal dominant hearing loss onset are post-lingual. Many genetic forms of mild-deafness are missed on many new born hearing screens and are picked up when a child starts school. These like the AD-deafness should not be considered “acquired” forms of deafness.

The authors need to use a more recent source that the Marazita ML et al 1993 study to estimate the frequency of acquired deafness. Over the past 27 year the understanding of the genetics of deafness and its epidemiology has drastically improved and this study might not be accurate.

Lines 142-146: Some of the terms the authors list cannot be used interchangeably. For example, inbreeding and assortative mating. These are not the same. One is based on genetic ancestry and the other is based on phenotype.

Roughly 6 generations (based on the authors calculation of 20year/generation) have passed since the Fay’s 1898 study. Has the rate of assortative mating among the deaf changed during this time?

How does the birth rate among hearing couple’s vs deaf-couples compare? Does this need to be considered?

Line 149: It is unclear where the 1.304% comes from. Please clarify. Please clarify the source of the allele frequencies.

Do these results apply to only outbred populations?

Line 266: delete “.”

Line 267-278: The evidence for GJB2-GJB6 digenetic inheritance is limited. It is now well established the large upstream deletions (some of which overlap GJB6) alter a cis-regulatory element of GJB2.

Line 273-278: The authors need to remove the hypothesis of a dominant digenetic GJB2/GJB6 allele. The hypothesis does not make sense and the biologically basis for it is questionable.

Line 285-293: Are the authors suggesting that haploinsufficiency of GJB2 is protective against dysentery OR that complete loss of GJB2 would be protective? It is unclear. If the latter, does that imply the rate of GJB2-related deafness would have increased during this time, given the survival advantage? It would be helpful for the authors to clarify this section.

Line 297: Please add citations

The rational of using the c.35delG, c.235delC and c.167delT variants only is confusing. It is also not clear why the authors use the frequency of these alleles only in the European population. The study might be more powerful if the authors used global population frequencies and use the frequencies for all loss of function alleles for GJB2.

Reviewer #2: In this study, the authors used an established forward-time genetics simulator to see whether intermarriage increases the prevalence of deafness alleles. I have the following comments:

Lines 166-167. What is “U”? is this Mann Whitney test statistics? Both the U value and p-value seems unusually high/low.

Lines 171-172. Similarly, why similar U values produce drastically different p-values?

Lines 167, 181, 217. Why the authors use 96% CI? This is very unconventional.

Did the authors simulated any random mating cases and compare the intermarriage simulation with the random mating simulation to see if there is any difference?

Minor:

Some references are not formatted correctly. See lines 163 and 196.

6. PLOS authors have the option to publish the peer review history of their article (what does this mean?). If published, this will include your full peer review and any attached files.

Reviewer #1: No

Reviewer #2: No

---

## [Author Response · Author response to Decision Letter 0]

30 Jul 2020

We included a Response to Reviewers as a separate file, per instructions.

---

## [Decision Letter · Decision Letter 1]

22 Sep 2020

PONE-D-20-07542R1

Deaf intermarriage does not increase the prevalence of deafness alleles

PLOS ONE

Dear Dr. Braun,

Thank you for submitting your manuscript to PLOS ONE. After careful consideration of reviewers' comments, we feel that the manuscript has greatly improved and addressed most of the comments. However, it does not fully meet PLOS ONE’s publication criteria as it currently stands. Therefore, we invite you to submit a revised version of the manuscript that addresses the points raised during the review process. The main concern is the use of an equation derived for inbreeding (similar genetic makeup) to estimate the outcomes of assortative mating (similar phenotype).  We recommend to carefully review and address the reviewer's comment and suggestions.

We look forward to receiving your revised manuscript.

Kind regards,

Hela Azaiez, Ph.D

Academic Editor

PLOS ONE

Reviewers' comments:

Reviewer's Responses to Questions

**Comments to the Author**

1. If the authors have adequately addressed your comments raised in a previous round of review and you feel that this manuscript is now acceptable for publication, you may indicate that here to bypass the “Comments to the Author” section, enter your conflict of interest statement in the “Confidential to Editor” section, and submit your "Accept" recommendation.

Reviewer #1: All comments have been addressed

Reviewer #2: All comments have been addressed

2. Is the manuscript technically sound, and do the data support the conclusions?

Reviewer #1: No

Reviewer #2: Yes

3. Has the statistical analysis been performed appropriately and rigorously? 

Reviewer #1: I Don't Know

Reviewer #2: Yes

4. Have the authors made all data underlying the findings in their manuscript fully available?

Reviewer #1: Yes

Reviewer #2: Yes

5. Is the manuscript presented in an intelligible fashion and written in standard English?

Reviewer #1: Yes

Reviewer #2: Yes

6. Review Comments to the Author

Reviewer #1: Line 283-286: This line is not correct. c.235delC is a founder mutation in the East Asians, c.167delT is a founder mutation in the Ashkenazi Jewish population and c.35delG is most common is Caucasians and white Europeans.

It is unclear how Fisher and Wright's mathematical model that was derived for inbreeding and assessing allele frequencies can be applied to phenotypes. With assertive mating, there is much variability at the genetic and allelic levels, however with inbreeding there is no new alleles being introduced. This needs to be further detailed. There is a big difference between allele frequency and phenotypic frequency, specifically for a recessive disease. This is why countries with a high degree of inbreeding have a much higher rate of recessive diseases.

A simpler way address what the authors want to is compare three simple scenarios:

1- The chance of having a deaf child if both parents are deaf due to the mutations in the same gene (ie 100% of their offspring will have hearing loss).

2- The chance of a deaf couple having a deaf child if both parents are deaf but different genes are responsible for their deafness.

3- The chance of a hearing couple having a deaf child.

Reviewer #2: (No Response)

7. PLOS authors have the option to publish the peer review history of their article (what does this mean?). If published, this will include your full peer review and any attached files.

Reviewer #1: No

Reviewer #2: No

---

## [Author Response · Author response to Decision Letter 1]

11 Oct 2020

The Response to Reviewers is included as a separate file, per instructions.

---

## [Decision Letter · Decision Letter 2]

14 Oct 2020

PONE-D-20-07542R2

Deaf intermarriage has limited effect on the prevalence of recessive deafness and no effect on underlying allelic frequency

PLOS ONE

Dear Dr. Braun,

Thank you for submitting your manuscript to PLOS ONE. After careful consideration, we feel that it has addressed all reviewers' comments and concerns. I only have one final  minor change to the last sentence of the conclusion " Therefore, Alexander Graham Bell’s fears of a “deaf variety of the human race” will not happen even if deaf intermarriage and assortative mating continue its present course." Although I understand what you are referring to, I believe replacing the word "fears" with concept or notion would be better perceived by the Deaf community and readers. we invite you to submit a revised version of the manuscript with the change requested above.

We look forward to receiving your revised manuscript.

Kind regards,

Hela Azaiez, Ph.D

Academic Editor

PLOS ONE

Reviewers' comments:

Reviewer's Responses to Questions

**Comments to the Author**

1. If the authors have adequately addressed your comments raised in a previous round of review and you feel that this manuscript is now acceptable for publication, you may indicate that here to bypass the “Comments to the Author” section, enter your conflict of interest statement in the “Confidential to Editor” section, and submit your "Accept" recommendation.

Reviewer #1: All comments have been addressed

2. Is the manuscript technically sound, and do the data support the conclusions?

Reviewer #1: Yes

3. Has the statistical analysis been performed appropriately and rigorously? 

Reviewer #1: Yes

4. Have the authors made all data underlying the findings in their manuscript fully available?

Reviewer #1: Yes

5. Is the manuscript presented in an intelligible fashion and written in standard English?

Reviewer #1: Yes

6. Review Comments to the Author

Reviewer #1: The authors have adequately addressed my comments.

7. PLOS authors have the option to publish the peer review history of their article (what does this mean?). If published, this will include your full peer review and any attached files.

Reviewer #1: No

---

## [Author Response · Author response to Decision Letter 2]

15 Oct 2020

We included a Response to Reviewers as a separate file, per instructions.

---

## [Editor Report · Decision Letter 3]

19 Oct 2020

Deaf intermarriage has limited effect on the prevalence of recessive deafness and no effect on underlying allelic frequency

PONE-D-20-07542R3

Dear Dr. Braun,

We’re pleased to inform you that your manuscript has been judged scientifically suitable for publication and will be formally accepted for publication once it meets all outstanding technical requirements.

Kind regards,

Hela Azaiez, Ph.D

Academic Editor

PLOS ONE
---

## [Editor Report · Acceptance letter]

23 Oct 2020

PONE-D-20-07542R3 

Deaf intermarriage has limited effect on the prevalence of recessive deafness and no effect on underlying allelic frequency 

Dear Dr. Braun:

I'm pleased to inform you that your manuscript has been deemed suitable for publication in PLOS ONE. Congratulations! Your manuscript is now with our production department. 

Kind regards, 

on behalf of

Dr. Hela Azaiez 

Academic Editor

PLOS ONE